# Cities in Competition: Is There a Link between Entrepreneurship and Development?

Zacharias Papanikolaou [1,*], Fani Kefala [2], Christos Karelakis [1], George Theodosiou [2] and Apostolos Goulas [3]

1 Department of Agricultural Development, Democritus University of Thrace, 68200 Orestiada, Greece
2 Department of Business Administration, University of Thessaly, 41500 Larissa, Greece
3 Department of Planning and Regional Development, University of Thessaly, 38221 Volos, Greece
* Correspondence: zpapanik@agro.duth.gr

**Abstract:** Cities operate in a competitive social environment requiring local authorities to adopt marketing strategies with significant economic ratings. City marketing that is related to the meaning of a city's name encourages activities in the city or region. The present study adopted a quantitative survey on a sample of 152 employees in companies to explore how important marketing is perceived for a city's development. The research was done in Trikala, a city in Greece. The key conclusion was that the more critical the participants consider the interventions in the city's natural environment, the more they believe that the city can benefit from corporate sponsorships. Subsequently, it appeared that the more they support the interventions in the structured environment of the city, the less they consider that corporate sponsorships can benefit it. It was explained that structured interventions usually involve very high investments that require funding from the central government, as sponsorships are not enough. The most substantial positive relationship was found between the importance of interventions in employment, entrepreneurship and tourism, and the importance of business sponsorships.

**Keywords:** city marketing; urban marketing; city brand name; territorial marketing

## 1. Introduction

In recent years there has been a growing recognition that cities are places that offer a wide range of functions and services available in the market for use. Cities also have a competitive business environment that requires local authorities to adopt marketing and branding strategies with significant financial implications. Therefore, entire cities and specific areas (e.g., city centers) seek to raise public awareness of their characteristics and image among different groups of users (i.e., residents, visitors, and employees). Nowadays, the whole world is a market. Can a citizen of one country do business with a citizen of another country? So, every country, city, and region is competitive as it shares consumers, tourists, investors, students, entrepreneurs, international sporting and artistic events, and many media with other governments and citizens of other countries. In recent years, many studies have been conducted on the strategies for managing the promotion of the city's image [1–4]. These studies agree that creating positive correlations and perceptions has a direct positive impact on the city. Positive perceptions and associations are people's views about a particular place. This makes the "brand name", i.e., the so-called "consumption" of the place, which is an essential tool for those who plan and implement development strategies [5]. This makes promoting the city or region's goods or services more accessible, and people pay more attention to entrepreneurship, education, technology, culture, etc. [6,7].

In recent years there has been a significant increase in the efforts made by city leaders, urban planners, and decision-makers worldwide to promote a positive and attractive image of their cities. These people believe that public images of their cities have far-reaching implications for important decisions made by:

- residents of other cities (to immigrate, visit or work);
- investors or managers of companies, industries, to move a factory, to start a business, to find business opportunities;
- the same residents of the city, whether they are going to stay or leave, if they will recommend the city to others, the value of the real estate, local pride;
- governments, whether budgets and resources will be allocated to industrial areas are created [8].

Due to growing competition between cities and increasing globalization—leading to more immigration, investment, and jobs outside the city—many cities worldwide are constantly "rebuilding" themselves and their resources to present an attractive image [8]. This is done to successfully compete with the international regime that could help attract tourists, conferences, sporting events, entrepreneurs, investors, businesses, headquarters, and global capital [8].

Considering those mentioned above, the present research refers to the importance of marketing and branding for developing a city and how quickly the competition in cities is evolving; competition in terms of how a city will become more attractive for accommodation, tourism, and investment, adding capital, trained staff, and technology. These three factors contribute to a city's economic development, region, or country. In this context, the opinion of the employees of the companies based in the Greek City of Trikala were investigated on the issue of the city's attractiveness and how these companies could contribute to the city's success. The aim of the research was to investigate and detect those factors that influence the marketing and branding of the city of Trikala.

Accordingly, the remainder of the paper involves a theoretical background regarding city marketing and branding in the next section, followed by the methods employed in the third section. Finally, the fourth section provides the study results, whereas the fifth section discusses the results and concludes.

## 2. City Marketing and Branding

Since the 1990s, most cities worldwide have been trying, many successfully, to change their image and economy, improving or redefining their characteristics and especially their cultural and economic profile [9]. Thus, the idea of the creative city became popular with the city's administrators (local government) [10]. Evans (2005) considers culture essential to making a city competitive, even though culture is recognized in urban politics or the quality of life. Thus, it lacks the criteria for measuring its rebirth. Florida [11] considers that strengthening the economy of a region or a city is done by attracting large companies and creating new jobs. Companies can relocate their headquarters or branch office to a city once they have identified the talent, skills, and competencies and whether there are tax breaks and political and job security. He also believes that to attract talent and create jobs, the city must provide infrastructure, highways, sports stadiums, shopping malls, etc. But if that happens, the city will become less attractive. A creative team or class should be developed that changes work, fun and daily life [12]. For cities to thrive, they must make full use of the talents and creativity of their inhabitants. Landry [5] considers cities the melting pot of culture and the creation points of wealth. Today, however, they face many problems, such as infrastructure, economic and social issues, and dramatic changes like the mass movement of desperate people from impoverished to rich western countries.

Miles & Paddison [13] argued that culture could be a driving force for urban economic development; it has become part of the ideology that cities seek to strengthen their competitive position. But what is remarkable is how governments and local development organizations have supported culture-oriented strategies to boost the urban economy and how the spread of strategies has globalized [10]. Cultural clusters have emerged as a new, alternative source of urban cultural development, blending cultural functions and activities from production to presentation and consumption, from theatre and visual arts to pop music and media, and are grouped into various spatial forms [14].

Another essential element for the economic development of the cities is the attractiveness of investments, which aims to attract new businesses. By the term investment attractiveness, we refer to a city's available resources and ability to maintain them, ensuring its development. It is a circular process as an attractive city attracts various social groups (tourists, students, investors, and new residents) [15]. According to the literature, the most critical factors shaping the attractiveness of a city are static and variable. Statics such as geographical location, available natural resources, and size cannot be replaced, whereas variables to be obtained must be created or configured. Examples of these are infrastructure, employment, education, and social services. In the literature, the factors that determine the investment attractiveness of a city are related to the estimation of production costs [16–18], such as transport costs, labor, and natural resource costs. Also, other factors that strongly influence the attractiveness of an area are the level of competition of the portal, the distance from suppliers, and the distance from public and private institutions (schools, banks, cultural centres, etc.).

The attractiveness of a city is directly related to its image as well as its identity. The concept of image is consumers' understanding of the overall activities of a business [19]. On the other hand, identity is defined as a set of associations created and maintained with a specific strategy [20]. Brand identity is how a company wants to present its brand to its audience [13], while brand image is its audience's interpretation of the brand identity [21]. According to the above, we could say that brand identity is the strategy that a company chooses to identify with its public, while the image is the brand's perception by the public [22].

### 2.1. Place Marketing

Two theoretical approaches are used when discussing site marketing and branding. One focuses on the success of blends of marketing strategies. The other one sees a relationship between marketing place and political economy, noting the collaboration of local government and businesses for growth in its economy [23]. The concept of place marketing, which some call "site promotion" or "city management", became popular in European literature in the 1980s and a little earlier in the United States [24]. There are many different definitions of "Place marketing" in the literature. According to Gold & Ward [2], it is "... the promotion of sites as the conscious use of publicity and marketing to communicate-disseminate selective images of specific geographical locations or areas to a targeted audience...". This definition distinguishes the use of options and desired images in the marketing process, its active role, and the target audience in the marketing plan's acceptance. Existing features or elements are selected and highlighted to make the place more appealing to that audience. This selectivity, however, often means ignoring or even hiding some of the negative features of the place [8].

### 2.2. The Marketing and Branding of Cities

The marketing and branding of cities have become critical issues of urban governance. In the global competition to attract tourists, residents and investment, cities apply the brand to develop an attractive image and a positive reputation. Almost all large and small cities implement strategies to improve their images [4]. City marketing and branding can be understood as the coordinated use of marketing tools supported by a common customer-oriented philosophy of creating, communicating, delivering, and exchanging urban offers of value to city customers and the city community in general. It refers to marketing tools to promote and develop regions, cities, and metropolitan areas. For this purpose, routine communication tools, such as advertisements and social media, are used to create brand communities, enhance the city's image by word of mouth, and activate positive relationships and perceptions of the city. However, place marketing involves more than site promotion [25]. It also includes policymaking for position improvement and public administration, such as attractive business tax policies. This means that policymakers adapt to the needs and desires of different target groups (from tourists to residents to foreign direct investment). In this sense, city marketing and branding are strategic planning

tools that parties can use to envision their future and support structural change in this direction. In the last decade, researchers and marketers have also faced the expansion of city branding, seeking to define it correctly to exploit its potential. Based on the international literature, the scholars came to a widely accepted definition of the city brand: an organized network based on a place's visual, verbal, and behavioral expression and its stakeholders. Zenker & Braun [26] consider the place brand complex because of the different target groups, the various locations offered, and the different organizations/companies that customers may have. Therefore, advanced branding management is required, including specific branding subcategories.

The international literature shows that most studies have a single purpose: establishing a link between place marketing, branding, and urban governance [27]. Boisen et al. [27] offered a holistic view of the conceptual difference between place promotion, place marketing and place branding. Their purpose is to resolve these concepts' ambiguity for practice and theory. Lucarelli [10] discusses the concept of city branding from its political dimension, indicating city branding as a hybrid form of urban politics where, for example, the boundaries between economics and politics, the market, and cities, are blurred and coexist. Finally, Braun et al. [28] dealt with the conflicts that arise from city branding and empirically tested two branding strategies: the first is an open process involving many stakeholders. The second is a branding approach that aligns the two schools of branding thinking (identity and image focus). They showed that both approaches lead to a city's higher reputation, but the open process shows more conflict.

In conclusion, the image of the city is formed according to the particular characteristics of each place combined with the human perspective [29] in contrast to city branding, which is a city's marketing strategy with the final purpose of promoting the place. The main characteristics of a city's branding are its image, uniqueness, and authenticity. In the effort of each city to redefine its image, it also tries to create its city branding [30]. Realizing the importance of city branding for the development of a place, this research comes to fill the gap regarding the investigation of the factors that influence the marketing and branding of the city of Trikala.

## 3. Study Area

Trikala is one of the four cities that form the Regional Unit of Trikala. Trikala, located in the middle of Greece (Figure 1), borders on the north with the City of Kalampaka, on the east with the Farkadona, on the southwest with Pyli and finally on the south with the city of Mouzaki, which, however, belongs to the Regional Unit of Karditsa. In addition, the Municipality of Trikala belongs to the largest Municipality in the country, having a total area of 608.48 km$^2$. According to the latest census of the Hellenic Statistical Authority (ELSTAT, 2011), the municipality of Trikala's permanent population amounts to 81,355 inhabitants, while the actual population is estimated at 80,287 inhabitants and the population of the city of Trikala, according to the 2011 census is 62,154 people. The local economy is based mainly on the agri-food sector as it is the region with the most significant production volume in dairy products and the first region in tsipouro standardization.

Furthermore, Trikala emerged as the first city in Greece that was considered digital in 2004 after the Municipality of Trikala decided to develop the city by turning it into "smart" [31,32]. The vision of the municipality was to upgrade the local economy, make proper use of energy and reduce the traffic problem, which is large enough for a provincial city like Trikala. Therefore, applications were created based on the use of information and communication technologies (ICT) to create a properly structured and organized digital city model, which will have as its primary goal the upgrade of the citizens' quality of life. Finally, it is one of the most important, as it possesses many historical monuments, buildings, churches, and museums [33].

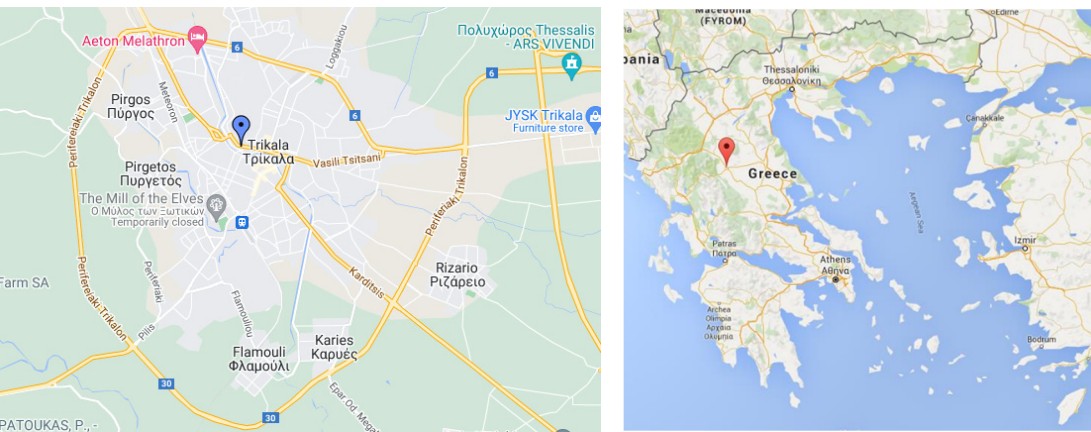

**Figure 1.** Topographic map of the study area [34].

## 4. Methodology

This study examined city marketing and branding qualitatively and quantitatively, employing a structured questionnaire with multiple-choice questions. First, the qualitative method of bibliographic review in the scientific field of Urban Marketing was implemented. Then, based on the theoretical findings, a quantitative questionnaire was created that was the main tool for collecting primary data from the study's population. Finally, the questionnaire was administered to a carefully selected population, which consisted exclusively of employees in companies that maintain their headquarters in the greater area of Trikala (Appendix A).

### 4.1. Data Sources and Collection

All the questions were based on an in-depth literature review regarding the questionnaire's content. The questionnaire questions included local city problems, interventions in the natural environment, the structured environment, quality of life, social policy, sports, culture, and entrepreneurship. Considering the area's rich natural environment, the development of the place, and the wealthy business activity, the authors chose these factors as the most important according to their perspective on the city of Trikala. The questionnaire was characterized by a clear structure, while the questions were grouped in solid sections based on their thematic content and graded on a scale of 0 to 5.

As for the data collection process, the survey instrument was administered to a carefully selected population, which consisted of employees in companies that maintain their headquarters in the greater area of Trikala. The purpose was to reach at least 250 people [35]. Therefore, all participants had to meet strict criteria for selection, such as working in locally based companies (small and medium enterprises in production, the marketing of products, and offering of business services). In choosing the final sample population, simple random sampling was adopted and accordingly, the questionnaire was sent to 250 people. Due to the constraints posed by the COVID-19 pandemic and the consequent exceptional restrictive measures imposed by the state authorities, the participants' sending, and subsequent completion, of the questionnaire were accomplished through a combination of email, which was collected with the help of the businesses and agencies of the city, and participants virtually completing using Skype and Zoom programs. The period for collecting the questionnaires was two months (October and November 2021). In the final stage, the conclusions of this study were derived from the quantitative analysis of the collected research data using the widespread statistical program IBM SPSS Statistics v.25. To achieve the maximum possible response rate, those respondents who did not respond with the first shipment were followed by two reminder emails, fifteen days and one month later. This systematic collection process eventually led to 152 fully completed questionnaires. This final sample size essentially represents a response rate of 60.2%, which is considered entirely satisfactory based on the relevant literature [36].

### 4.2. Research Method

Descriptive statistics were employed for the percentage representation of the independent variables. In addition, frequency analyses and One-Way Analysis of Variance (ANOVA) were performed to check whether the samples differed due to the effect of a single factor along with multiple linear regression analysis. The purpose of the multiple regression analysis was to describe the relationship between the dependent variable Y and the k independent variables $X_1$, $X_2$, D, and $X_k$. This model has the form:

$$\Upsilon = \beta_0 + \beta_1 \times X_1 + \beta_2 \times X_2 + \ldots + \beta_\kappa \times X_k + \varepsilon,$$

where $\beta_0$: a constant, $\beta_1$, $\beta_2$, ... , $\beta_\kappa$: the regression coefficients that describe the effect of the independent variables, $\varepsilon$: error.

The regression coefficients (otherwise known as the slope parameters) determine the change in the expected value of Y in each unit change of Xi when all other variables are kept constant. In our study, the dependent variables were threefold: whether the city can be helped with the companies' sponsorships; whether the area's development can contribute to the development of the participant's business; and sales percentage. In contrast, the independent variables involved the natural environment, structured environment, quality of life, social politics, sports and education, civilization, and employment tourism business.

### 5. Results

Table 1 shows the mean values and standard deviations of the participants' answers to each of the questions included in the questionnaire of the present survey.

**Table 1.** Descriptive statistical analysis (Mean Values, Standard Deviations) for the Research Questionnaire Questions.

| | Mean Values | Standard Deviations |
|---|---|---|
| Most Important Local Problems of the City | | |
| Road and Rail Direct Connection to the City | 3.03 | 1.209 |
| No easy Access to the Pindos Mountain Range | 2.7 | 1.047 |
| Cooperation of the City with the University and local bodies | 2.68 | 1.083 |
| Introversion to culture, tourism, education, and investment | 2.73 | 1.156 |
| The environmental pollution | 3.34 | 1.049 |
| Accesses for People with Special Needs | 3.57 | 1.001 |
| Interventions in the Natural Environment | | |
| Green (parks, etc.) | 3.39 | 1.062 |
| Aquatic Environment | 3.29 | 1.027 |
| City noise | 2.95 | 1.192 |
| Fauna (the whole animal kingdom) | 3.24 | 0.935 |
| Interventions in the Structured Environment | | |
| Urban planning | 3.24 | 0.919 |
| Land Registry | 3.03 | 1.01 |
| Building Infrastructure | 3.28 | 0.957 |
| Bicycle paths | 3.42 | 0.987 |

**Table 1.** *Cont.*

|  | Mean Values | Standard Deviations |
|---|---|---|
| **Interventions in Quality of Life** | | |
| City Cleaning and Garbage Management | 3.66 | 1.042 |
| Maintenance and Increase of Public Green Spaces | 3.52 | 0.92 |
| Road Transport and Connection with other cities | 3.23 | 0.945 |
| Construction and Operation of a Regional Airport | 3.16 | 1.136 |
| **Interventions in Social Policy** | | |
| Care for Preschool Children and Public Kindergartens | 3.38 | 0.969 |
| Modernization of School Units | 3.29 | 0.896 |
| Public Sports Venues | 3.34 | 0.977 |
| Preventive Medicine for Residents, Old Age, Disabled | 3.23 | 1.013 |
| **Interventions in Sports—Education** | | |
| Sports Programs for Children, Women, Men, and the Disabled | 3.2 | 1.044 |
| Educational Programs for the timeless culture of the city | 2.74 | 0.954 |
| Educational Programs for the consumption of local products, food, and food hygiene | 2.97 | 1.051 |
| **Interventions in Culture** | | |
| Utilization of the music tradition | 3.02 | 0.88 |
| Establishment and Utilization of electronic and non-library | 2.92 | 0.865 |
| Cultural events in collaboration with other municipalities of the Region | 2.97 | 0.792 |
| Highlighting the historical elements of the city | 3.09 | 1.054 |
| **Interventions in Entrepreneurship-Employment-Tourism** | | |
| Foreign language education programs for adults | 2.9 | 1.078 |
| Marketing and entrepreneurship training programs | 2.7 | 0.942 |
| Establishment of an office to find national and international markets and products | 2.91 | 0.976 |
| Finding places for the operation of start-ups | 2.68 | 1.045 |
| Collaborations with municipalities of the Region of Thessaly and other Regions | 2.91 | 1.035 |
| **Company Participation** | | |
| Do you think that the city can be helped with the sponsorships of the companies | 2.85 | 1.082 |
| Your business was/is a sponsor of an event or events of the city | 2.78 | 1.051 |
| The development of the area can contribute to the development of your business | 3.23 | 1.047 |
| Do you think the business's development can take place through the voluntary branded offer of your business? | 3.12 | 1.137 |
| Percentage of sales you could offer as sponsorship of one or more City activities? | 1.78 | 0.824 |

The one-way analysis of variance—ANOVA (Table 2) analysis confirmed that there are several statistically significant differences in the answers of respondents belonging to companies in different industries and at a significance level of 5%. It was found that respondents in the production and marketing sector believe to a greater extent that the city can be helped by business sponsorships (mean value = 3.20), compared to service

companies and purely commercial companies, at 2.4 and 2.91, respectively. Subsequently, the productive enterprises tend to sponsor the city's events (mean value = 3.05) compared to the rest (average prices 2.38 and 2.89, respectively). Going forward, the productive enterprises embrace to a greater extent the fact that the development of the wider area is expected to benefit them as well (mean value = 3.71). In contrast, service companies (average price = 2.64) and commercial companies (mean value = 3.24) do not firmly believe in this view. In the same pattern, and in terms of whether the growth can take place through the voluntary branded offer of each company, again, the productive companies stand out (mean value = 3.51), with the purely commercial ones coming in second place (mean value = 3.24) and the latest service companies (mean value = 2.58). Subsequently, the same trend is maintained in terms of the percentage of their sales, which would be willing to channel sponsorships to the city, with production companies ahead of the rest. Companies, regardless of industry, do not seem inclined to devote more than 1% of their turnover to such actions. Turning now to the possible activities that the city could undertake in many sectors, again, the productive enterprises seem to applaud to a greater extent (mean value = 3.60) the necessary interventions of the city in the structured environment of the area (urban planning, cadastre, building infrastructure, bicycle paths) compared to both service providers (mean value = 2.83) and purely commercial enterprises (average price = 3.24). Regarding all other categories of interventions (natural environment, quality of life, social policy, sports—education, culture, and employment-entrepreneurship-tourism,) no statistically significant differences were found in the responses of different types of companies at the level of statistical significance of 5%.

**Table 2.** One-Way Analysis of Variance (ANOVA) with Criterion of the Business Activity.

| | Business Activity | N | Mean | Std. Deviation | F | Sig. |
|---|---|---|---|---|---|---|
| Do you think that the municipality can be helped with the sponsorships of the companies | Production and Marketing of Products | 55 | 3.2 | 1.297 | | |
| | Offer of Services | 50 | 2.4 | 0.571 | 7.96 | 0.001 |
| | Product Marketing | 47 | 2.91 | 1.071 | | |
| | Total | 151 | 2.85 | 1.082 | | |
| Your business was/is a sponsor of an event or events of the municipality | Production and Marketing of Products | 55 | 3.05 | 1.367 | | |
| | Offer of Services | 50 | 2.38 | 0.635 | 6.147 | 0.003 |
| | Product Marketing | 47 | 2.89 | 0.849 | | |
| | Total | 151 | 2.78 | 1.051 | | |
| The development of the area can contribute to the development of your business | Production and Marketing of Products | 55 | 3.71 | 1.149 | | |
| | Offer of Services | 50 | 2.64 | 0.693 | 16.628 | 0.000 |
| | Product Marketing | 47 | 3.28 | 0.935 | | |
| | Total | 151 | 3.23 | 1.047 | | |
| Do you think the business's development can take place through the voluntary branded offer of your business? | Production and Marketing of Products | 55 | 3.51 | 1.386 | | |
| | Offer of Services | 50 | 2.58 | 0.758 | 10.237 | 0.000 |
| | Product Marketing | 47 | 3.24 | 0.923 | | |
| | Total | 151 | 3.12 | 1.137 | | |
| Percentage of Sales that can be allocated as a Sponsorship of one or more activities of the municipality | Production and Marketing of Products | 55 | 2.07 | 0.79 | | |
| | Offer of Services | 50 | 1.28 | 0.809 | 17.006 | 0.000 |
| | Product Marketing | 47 | 1.98 | 0.614 | | |
| | Total | 151 | 1.78 | 0.824 | | |

**Table 2.** *Cont.*

| | Business Activity | N | Mean | Std. Deviation | F | Sig. |
|---|---|---|---|---|---|---|
| Natural environment | Production and Marketing of Products | 55 | 3.35 | 1.14423 | 2.067 | 0.130 |
| | Offer of Services | 50 | 3.005 | 0.63183 | | |
| | Product Marketing | 47 | 3.2979 | 0.89479 | | |
| | Total | 152 | 3.2204 | 0.93006 | | |
| Structured Environment | Production and Marketing of Products | 55 | 3.6091 | 0.93512 | 12.184 | 0.000 |
| | Offer of Services | 50 | 2.83 | 0.73269 | | |
| | Product Marketing | 47 | 3.2447 | 0.71758 | | |
| | Total | 152 | 3.2401 | 0.86549 | | |
| Quality of life | Production and Marketing of Products | 55 | 3.5545 | 1.02593 | 1.791 | 0.170 |
| | Offer of Services | 50 | 3.24 | 0.5691 | | |
| | Product Marketing | 47 | 3.3723 | 0.89058 | | |
| | Total | 152 | 3.3947 | 0.86052 | | |
| Social politics | Production and Marketing of Products | 55 | 3.4136 | 0.97565 | 2.222 | 0.112 |
| | Offer of Services | 50 | 3.115 | 0.67803 | | |
| | Product Marketing | 47 | 3.3989 | 0.69286 | | |
| | Total | 152 | 3.3109 | 0.8092 | | |
| Sports—Education | Production and Marketing of Products | 55 | 3.0545 | 1.12905 | 0.775 | 0.463 |
| | Offer of Services | 50 | 2.84 | 0.86042 | | |
| | Product Marketing | 47 | 3.0071 | 0.66117 | | |
| | Total | 152 | 2.9693 | 0.91537 | | |
| Civilization | Production and Marketing of Products | 55 | 3.1 | 0.90344 | 1.189 | 0.307 |
| | Offer of Services | 50 | 2.885 | 0.60443 | | |
| | Product Marketing | 47 | 3 | 0.54921 | | |
| | Total | 152 | 2.9984 | 0.71438 | | |
| Employment-Entrepreneurship-Tourism | Production and Marketing of Products | 55 | 3.0218 | 1.08435 | 2.339 | 0.100 |
| | Offer of Services | 50 | 2.648 | 0.9377 | | |
| | Product Marketing | 47 | 2.7957 | 0.51834 | | |
| | Total | 152 | 2.8289 | 0.90133 | | |

In the last stage of the statistical analysis, a linear multiple regression analysis was performed with a dependent variable to the extent that participants believe that the city can be helped with business sponsorships and independent variables to the extent that participants believe to be implemented in the city's natural and structured environment, in the direction of the quality of life, social policy, sports—education, culture and employment-entrepreneurship-tourism. Initially, the value of the Adjusted $R^2$ index certifies that, without a doubt, the independent variables selected are responsible for 45.2% of the variance of the dependent variable. The result, combined with the value and the statistical significance of the F index (18,697, $p = 0.000$), justifies the choice of the specific dependent variables concerning the particular independent variables. The regression method used was Forward. Table 3 shows the detailed results of the regression analysis. Five statistically significant correlations were identified between the degree to which participants believe that business sponsorships can help the city and the importance with which they considered the implementation of interventions in: (a) the natural environment; (b) the structured environment; (c) sport and education; (d) culture; and (e) employment-entrepreneurship-tourism.

Table 3. Multiple Linear Regression Analysis Results.

| Dependent Variable: | The Municipality Can Be Helped with the Sponsorships of the Companies | | The Development of the Area Can Contribute to the Development of Your Business | | The Sales Percentage | |
|---|---|---|---|---|---|---|
| Variables | B | *p* | B | *p* | B | *p* |
| Natural environment | 0.392 | 0 | 0.113 | 0.223 | 0.249 | 0.014 |
| Structured Environment | −0.381 | 0.001 | −0.075 | 0.532 | 0.218 | 0.098 |
| Quality of life | 0.229 | 0.177 | 0.184 | 0.303 | −0.323 | 0.099 |
| Social politics | 0.174 | 0.384 | 0.122 | 0.563 | 0.449 | 0.053 |
| Sports and Education | 0.25 | 0.023 | 0.19 | 0.101 | −0.144 | 0.251 |
| civilization | −0.435 | 0.008 | −0.583 | 0.001 | −0.299 | 0.111 |
| Employment Tourism business | 0.518 | 0 | 0.741 | 0 | 0.387 | 0.002 |
| | $R^2 = 0.478$. Adjusted $R^2 = 0.452$. F = 18.697 | | $R^2 = 0.418$. Adjusted $R^2 = 0.389$. F = 14.659 | | $R^2 = 0.308$. Adjusted $R^2 = 0.74$. F = 9.09 | |

Furthermore, a strong positive and statistically significant correlation was found between the dependent variables and the importance of interventions in the natural environment (b = 0.392, *p* = 0.00). In simple words, the more critical the participants consider the interventions in the city's natural environment, the more they believe the city can benefit from the corporate sponsorships. Subsequently, a negative and statistically significant correlation was found (b = −0.381, *p* = 0.001) between the dependent variables and the importance of possible municipal interventions in the structured environment.

The more they supported such interventions, the less they thought corporate sponsorships could be of benefit. This may be explained because structured interventions usually involve very high investments that require funding from the central government, as sponsorships are not enough.

In the next step, a linear multiple regression analysis was performed with a dependent variable on the extent to which the participants believe the area's development can contribute to their business development. Initially, the value of the Adjusted $R^2$ index certifies that, without a doubt, the independent variables selected are responsible for 38.9% of the variance of the dependent variable. This result, combined with the value and the statistical significance of the F index (14,659, *p* = 0.000), justifies the choice of the specific dependent variable in relation to the particular independent variables. Only two statistically significant correlations were identified between the extent to which participants believed that the area's development can contribute to the development of their business and the importance of interventions that the city may implement in the fields of culture and employment-entrepreneurship-tourism. A statistically significant and negative correlation was recorded between the importance of interventions in the field of culture and the degree to which participants believe that the region's development can also help develop their business (b = −0.583, *p* = 0.001).

On the contrary, a very strong positive and statistically significant relationship was found between the extent to which the region's development can benefit businesses and the importance of interventions in employment, entrepreneurship, and tourism (b = 0.741, *p* = 0.000). The respondents believe the interventions enhancing the region's economic activity will benefit the region and local businesses. No statistically significant correlations were found for all other independent variables with the dependent variable.

Finally, a linear multiple regression analysis was performed with a dependent variable, the percentage of sales the participants would be willing to offer as sponsorship of one or more city actions. Initially, the value of the Adjusted $R^2$ index certifies that, without a doubt, the independent variables selected are responsible for 27.4% of the variance of the dependent variable. This result, combined with the value and the statistical significance of the F index (9.09, *p* = 0.000), justifies the choice of the specific dependent variable in

relation to the specific independent variables. Two statistically significant correlations were identified between the percentage of sales of their businesses that the participants would be willing to allocate as sponsorship of one or more activities of the city and the importance they consider the implementation of interventions on the one hand in the natural environment and on the other of the employment-entrepreneurship-tourism. Furthermore, a positive and statistically significant correlation was found between the sales percentage and the importance of interventions in the natural environment (b = 0.249, *p* = 0.014). In simple words, the more critical the participants consider the interventions in the city's natural environment, the higher percentage of their sales they are willing to offer as sponsorship of one or more city activities. Subsequently, a strong positive and statistically significant correlation was found (b = 0.387, *p* = 0.002) between the dependent and the importance of possible municipal interventions in employment, entrepreneurship, and tourism. In simple words, this means that the participants consider these interventions more important, the higher percentage of their sales they are willing to offer in the form of sponsorship to support the City's actions. No statistically significant correlations were found for all other independent variables with the dependent variable.

## 6. Discussion

As it turns out, in the modern era characterized by globalization, cities are one of the most important economic centres. Competition between nations has now shifted to competition between cities [37]. The goal of cities now is to attract investors. They compete to be as accessible as possible to new business activities [11]. However, the problems they must face are many. One is the weak image of small towns that comes from size and location. These two characteristics are mainly the cause of the small number of visitors. The lack of personal contact between residents and visitors creates stereotypes that also prevent the lack of investors [38]. The most important local problems faced by the city of Trikala involved access for people with disabilities, environmental pollution and the direct road and rail connection of the city with the rest of Greece and the mainland. In contrast to Athens, where the main problems have to do with the architectural design approach [39], in Italian cities, environmental sustainability is one of the most important problems [17].

Then, regarding the interventions that should be made regarding the city's natural environment, respondents believed that the most critical aspect relied on green spaces (parks, etc.) and the aquatic environment of the city. Continuing, in terms of interventions necessary for the city's structured environment, the emphasis seems to be on creating bike lanes, followed by new building infrastructure, and promoting urban planning. In addition, in what has to do with the necessary interventions to improve the citizens' quality of life, the participants pointed out the improvement of cleanliness and waste management and the maintenance and increase of public green spaces more urgent. Regarding social policy interventions, respondents believed that an essential aspect relied on caring for preschool children and strengthening public kindergartens, followed by creating new public sports facilities. In the next stage, in sports and education, the respondents indicated creating sports programs for children, women, men, and the disabled is an absolute priority. In contrast, creating educational programs aimed at promoting local products and food consumption was considered very important as food hygiene and the timeless culture of the city. Finally, it is worth emphasizing that the education system's modernization is a crucial priority for the EU so that citizens can gradually succeed in a complex world facing rapid technological, economic, and cultural change [40].

In the next stage, the need to highlight the historical elements of the city and the condition to utilize the musical tradition of the area was emphasized. At this point, it is worth emphasizing that Trikala is rich in medieval monuments. For example, the middle of the old city is characteristic of the castle of Trikala, with an ottoman clock tower being the typical icon of the city, also known as Trikala's Eiffel Tower [40]. Then, in terms of possible interventions to be implemented at the level of business and tourism activities with emphasis on employment, entrepreneurship and tourism, the findings

showed that respondents believed that the most important aspect relied mainly upon three areas: foreign language education programs for adults; the establishment of an office to find national and international markets and products; and collaborations with municipalities of the neighboring region of Thessaly and others. Finally, it seemed that the participants believed that the area's development could contribute to their business and that the city's development could occur through the voluntary offer of their own business.

More specifically, it was found that companies in the production and marketing sector believe to a greater extent, that the city can be helped by business sponsorships against service companies and purely commercial companies. On the other hand, service and commercial companies do not seem to share this view to such an extent. Then, regarding whether growth can occur through each company's voluntary branded offer, the productive companies stand out, with the purely commercial and service companies following in second and third place, respectively. The same trend is maintained regarding the percentage of their sales, which would be willing to channel sponsorships to the city, with productive companies ahead of the rest. Companies, regardless of industry, do not seem inclined to devote more than 1% of their turnover to such actions. In the last stage of our analysis, we performed linear multiple regression analyses to identify significant interdependencies between our research variables.

The results of the study raise some interesting issues. Initially, it was found that the more important the participants considered the interventions in the city's natural environment, the more they believed the city could benefit from the corporate sponsorships. In addition, as expected, the most substantial positive relationship was found between the importance of interventions in employment, entrepreneurship and tourism and the importance attached to business sponsorships. Besides, the tourism interest of the city is well known, which helps in the rural development of the area. Moreover, one of the biggest attractions is Meteora, a Natural World Heritage Monument protected by UNESCO [41].

Furthermore, the city's cultural development can positively impact local businesses. Still, the respondents argued that the interventions enhancing the region's economic activity would benefit the region and, ultimately, the local businesses. Subsequently, the finding was recorded that the more critical the participants consider the interventions in the city's natural environment, the higher percentage of their sales they are willing to offer as sponsorship of one or more city activities. Moreover, the more critical the participants consider the interventions in the fields of employment, entrepreneurship and tourism, the higher percentage of their sales they are willing to offer in the form of sponsorship to support the city's actions.

## 7. Conclusions

In conclusion, setting the initial goal is the investigation of the factors that influence the marketing of the city of Trikala. The most significant results of the study indicate that the more critical the participants consider the interventions in the city's natural environment, the more they believe that it can benefit from the corporate sponsorships, with essential priorities in the green spaces and the city's aquatic environment. Subsequently, it appeared that the more they support the interventions in the structured environment of the city, the less they consider that corporate sponsorships can benefit the city. This is explained by the fact that structured interventions usually involve very high investments that require funding from the central government, as sponsorships are not enough. Typical examples are the need for large projects, such as bike paths and new building structures. The most substantial positive relationship between the importance of interventions in employment, entrepreneurship and tourism and the importance attached to business sponsorships, mainly on adult education and collaborations with other bodies. Some limitations of the research were that it was conducted in a specific city with specific characteristics, a small sample, and it included only employees. Future research could be done in corresponding small and large cities abroad to compare and confirm the results.

**Author Contributions:** Conceptualization, G.T.; Data curation, Z.P. and C.K.; Formal analysis, Z.P.; Investigation, F.K. and A.G.; Methodology, Z.P. and C.K.; Project administration, F.K., G.T. and A.G.; Software, Z.P., F.K. and G.T.; Supervision, C.K. and G.T.; Writing—original draft, F.K. and A.G.; Writing—review & editing, Z.P., C.K. and G.T. All authors have read and agreed to the published version of the manuscript.

**Funding:** This research received no external funding.

**Institutional Review Board Statement:** Not applicable.

**Informed Consent Statement:** Not applicable.

**Data Availability Statement:** Not applicable.

**Conflicts of Interest:** The authors declare no conflict of interest.

**Appendix A**

1. What do you think are the most important local problems of the City?
Please write them down and mark next to them the degree of importance from 1 to 5.
1 = not important at all 2 = relatively important 3 = important 4 = quite important
5 = very important

| 1 = Not at all important 2 = Relatively important 3 = Important 4 = Fairly important 5 = Very important | | | | | | |
|---|---|---|---|---|---|---|
| 1º | Road and rail direct connection of the City with the rest of Greece. | 1 | 2 | 3 | 4 | 5 |
| 2º | The non-existent train as a means of transporting people and goods in large urban centres and abroad. | 1 | 2 | 3 | 4 | 5 |
| 3º | The difficult Access to the mountain massif of Pindos in winter due to the poor road network. | 1 | 2 | 3 | 4 | 5 |
| 4º | The bad road network of the City and the prefecture in general. | 1 | 2 | 3 | 4 | 5 |
| 5º | The municipality's cooperation with the University of Thessaly for the municipality's development. | 1 | 2 | 3 | 4 | 5 |
| 6º | The cooperation of the municipality with local and regional bodies | 1 | 2 | 3 | 4 | 5 |
| 7º | The introversion of the City in matters of culture, tourism, education and investment. | 1 | 2 | 3 | 4 | 5 |
| 8º | The non-existence of places of culture and cultural events. | 1 | 2 | 3 | 4 | 5 |
| 9º | Lack of public green spaces | 1 | 2 | 3 | 4 | 5 |
| 10º | The environmental pollution | 1 | 2 | 3 | 4 | 5 |
| 11º | The development of the City by height and not by area. | 1 | 2 | 3 | 4 | 5 |
| 12º | City security. | 1 | 2 | 3 | 4 | 5 |
| 13º | Access for people with special needs | 1 | 2 | 3 | 4 | 5 |
| 14º | Lack of full online service for businesses and citizens with the public and financial sector (banks). | 1 | 2 | 3 | 4 | 5 |

2. In your opinion, what interventions (projects) should be done to improve the general situation of the municipality throughout its area? (after Kallikrates):

| 1. Natural environment | | | | | | |
|---|---|---|---|---|---|---|
| 1 = not important 2 = less important 3 = important 4 = fairly important, 5 = very important | | | | | | |
| 1º | Green (Parks, etc.) | 1 | 2 | 3 | 4 | 5 |
| 2º | Aquatic environment | 1 | 2 | 3 | 4 | 5 |
| 3º | Air pollution | 1 | 2 | 3 | 4 | 5 |
| 4º | City noise | 1 | 2 | 3 | 4 | 5 |
| 5º | Fauna (the totality of the animal kingdom of a geographical area) | 1 | 2 | 3 | 4 | 5 |
| 2. Structured environment | | | | | | |
| 1 = not important 2 = less important 3 = important 4 = fairly important, 5 = very important | | | | | | |
| 1º | Urban planning | 1 | 2 | 3 | 4 | 5 |
| 2º | Land registry | 1 | 2 | 3 | 4 | 5 |
| 3º | Building infrastructure | 1 | 2 | 3 | 4 | 5 |
| 4º | Streets-parking lots-sidewalks-walkways | 1 | 2 | 3 | 4 | 5 |
| 5º | Bike lanes | 1 | 2 | 3 | 4 | 5 |

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
