# Peer review of "Cities in Competition: Is There a Link between Entrepreneurship and Development?"

_world, doi:10.3390/world3040051_

Round 1

Reviewer 1 Report

The objective stated at page 2 ("how important marketing is for developing a city and how fast the competition of cities is evolving") cannot be a research objective since the first part is very general and it was demonstrated already, and the second could not be fulfilled by this paper. Following, the research questions stated at page 2 are not correct, since they do not need an academic research and they are simple city management questions (might be the objectives of a satisfaction survey initiated by the Mayor of the city). 

The conceptual frame is unclear in the paper. There is no clear distinction between city image and city branding, as the authors refer to them (in spite of the fact that they are distinct concepts). Also the references to important sources in the international scholarship are missing. The literature review is very vague and readers cannot understand which are the gaps that the authors intend to resolve. 

Further, the methodology brings more confusion. First of all, the qualitative research claimed in the abstract reveals to be only the usual bibliographical review performed for papers (p.5). To be a method in itself, the bibliographical study has to meet several criteria and to follow some clear academic rules, but it is not the case here.

Further, the design of research is very unclear. Nothing is said about the variables (why these variables were considered and no others), about the structure of the instrument (sections, items, scales). Also, which were the criteria of selection for the sample and why the respondents were considered adequate, and which kind of local companies were selected (field, size). An important question here is how the responses of the employees of local companies are significant for the  will to offer a sponsorship for city actions since they are not involved in the companies' management. Also, why their perception that their company's development is influenced by the development of the area is relevant, since they are not in management positions. 

Further, several wrong statements are made, e.g. p.6, "companies in the production and marketing sector believe to a greater extent that the city can be helped by business sponsorships": respondents believe, and not the companies. Also, wrong expressions are used in this section, e.g. "average price"(??) instead of "mean value".

Also, interpretations of results are wrong in some points, e.g. p.11 "the participants do not consider that the city's cultural development can positively impact their business" - the correlation of the two variables in table 3 is about beliefs & importance of intervention in the cultural sector, nothing about the impact of city's cultural development on their own company. 

Further, instead of discussing the findings and comparing them with the findings of previous studies, the discussion section is proposing some policy measures appropriate for city management. Finally, conclusion does not return to the paper objectives in order to place the most important outcomes of research, as it should be made in the final part of a paper. 

Serios spelling is needed, e.g. letters missing in the subtitles, and sometimes words are missing (e.g. p.12 "significant correlations was found between the dependent and the importance of....").

Reviewer 2 Report

The aim of this article is to explore how important marketing is for developing a city and how fast the competition of cities is evolving. The paper contributes to the existing literature and knowledge on city branding. However, the authors are invited to revise the manuscript to address specific concerns before a final decision is reached. The following are needed in the paper if it is to be resubmitted:

·       The information provided in the abstract is somewhat misleading. The research done is only quantitative and not both quantitative and qualitative. The sample is 152 people and not 250 as stated. The sample consists of employees in companies and not just people. It should also be mentioned that the research was done in Trikala, a city in Greece.

·        The authors need to write purpose of the paper in a way that will justify the unique contribution of this paper. The originality statement of the article should be clearly articulated by supporting more theoretical clarification/evidence as well as rationale why should the reader be interested in.

·       At various points in the article (e.g. lines 35, 69…) the authors only refer to marketing and marketing strategies. In my opinion branding is more important here and should be mentioned along with marketing.

·       In the Introduction, line 73, please change the phrase “the opinion of the companies…” with “the opinion of the employees of the companies…”.

·       Authors should change the title of Section 2 to “City Marketing and Branding”.

·       Authors should enrich the literature review. In the first part they should answer the question “What makes a city attractive?”. The authors focus mainly on culture but there are many more such factors. They should also explain how these factors differ in the different groups of people mentioned (e.g. residents, visitors, workers, investors, students, etc.). The literature review on city branding also needs to be further strengthened. For example, reference to the concepts of city image and city identity are important. The meaning of the city image should be clarified, the differences between the city image and the city identity should be explained, and the importance of the city image for the success and the attractiveness of the city should be highlighted. The alignment between these two (identity and image) is the key for a successful city branding strategy.

·       In Section 3, line 175, 81,355 inhabitants are the population of the Municipality of Trikala and not of the city of Trikala as stated. The population of the city according to the 2011 census is 62,154 people. Please correct this.

·       In Figure 1 please add source.

·       Please, correct the title in section 4 to "Methodology".

·        Some critical information is missing from the methodology section. Authors must indicate when the research took place (time period). They must explain how they found the e-mail addresses of the participants in order to send them the questionnaire. Why were Zoom/Skype meetings conducted to complete the questionnaires? Weren't they self-completion? In which statistical program was the statistical analysis performed?

·        In the results section, add a table with the demographic characteristics of the sample (e.g. gender, age, marital status, education, income), if there were such questions.

·       The text in the discussion section should be broken into paragraphs because it is hard to read.

·        In the discussion section, there is simply a repetition of the main results of the research. Here, a correlation with the findings of previous research is required.

·        In the Conclusion Section, please add some limitations of the research (e.g. the research was conducted in a specific city with specific characteristics, the sample was small, it included only employees etc.) and some directions for future research.

The list of references needs more editing.

Reviewer 3 Report

-The state-of-the-art section seems a little too broad concerning the specific research questions of the research.

-Section on methods needs to be enhanced. Which are the "strict" criteria for someone to be part of the sampling frame? Furthermore, how many people were finally part of the sampling frame.

-An empty sample of the questionnaire will be helpful. On what scale are the questions graded, it was 0 to 5, 0 to 7..0 to 10?

-Discussion should be shorter and more meaningful.

Round 2

Reviewer 1 Report

The objectives of research are still unclear (should be stated in introduction or literature review). In the abstract is mentioned as objective.... "[to explore] how important marketing is for developing a city and how fast the competition of cities is evolving]. Correct is ...." to explore how important is marketing perceived for a city's development....)

At section 2.1 the correct title is "Place marketing" not "Marketing place"

Discussion is still too long and should adopt a more balanced style (e.g. not "priority should be given" but "respondents believed that. the most important aspect relied in...)

Reviewer 2 Report

The reviewer's comments have been addressed. The text has been improved in quality. The list of references needs more editing. A common way of citing references should be adopted according to journal instructions.
